# Science to Commerce: A Commercial-Scale Protocol for Carbon Trading Applied to a 28-Year Record of Forest Carbon Monitoring at the Harvard Forest

Nahuel Bautista [1], Bruno D. V. Marino [2,*] and J. William Munger [3]

[1] Planetary Emissions Management Inc., Cambridge, MA 02139, USA; nahuel.bautista@pem-carbon.com
[2] Executive Management, Planetary Emissions Management Inc., Cambridge, MA 02139, USA
[3] Department of Earth and Planetary Sciences, School of Engineering and Applied Sciences, Harvard University, Cambridge, MA 02139, USA; jwmunger@seas.harvard.edu
[*] Correspondence: bruno.marino@pem-carbon.com

**Abstract:** Forest carbon sequestration offset protocols have been employed for more than 20 years with limited success in slowing deforestation and increasing forest carbon trading volume. Direct measurement of forest carbon flux improves quantification for trading but has not been applied to forest carbon research projects with more than 600 site installations worldwide. In this study, we apply carbon accounting methods, scaling hours to decades to 28-years of scientific $CO_2$ eddy covariance data for the Harvard Forest (US-Ha1), located in central Massachusetts, USA and establishing commercial carbon trading protocols and applications for similar sites. We illustrate and explain transactions of high-frequency direct measurement for $CO_2$ net ecosystem exchange (NEE, gC m$^{-2}$ year$^{-1}$) that track and monetize ecosystem carbon dynamics in contrast to approaches that rely on forest mensuration and growth models. NEE, based on eddy covariance methodology, quantifies loss of $CO_2$ by ecosystem respiration accounted for as an unavoidable debit to net carbon sequestration. Retrospective analysis of the US-Ha1 NEE times series including carbon pricing, interval analysis, and ton-year exit accounting and revenue scenarios inform entrepreneur, investor, and landowner forest carbon commercialization strategies. $CO_2$ efflux accounts for ~45% of the US-Ha1 NEE, an error of ~466% if excluded; however, the decades-old coupled human and natural system remains a financially viable net carbon sink. We introduce isoflux NEE for t$^{13}$C$^{16}$O$_2$ and t$^{12}$C$^{18}$O$^{16}$O to directly partition and quantify daytime ecosystem respiration and photosynthesis, creating new soil carbon commerce applications and derivative products in contrast to undifferentiated bulk soil carbon pool approaches. Eddy covariance NEE methods harmonize and standardize carbon commerce across diverse forest applications including, a New England, USA regional eddy covariance network, the Paris Agreement, and related climate mitigation platforms.

**Keywords:** commercial eddy covariance; daytime ecosystem respiration; forest carbon sequestration; Harvard Forest; NEE; t$^{13}$C$^{16}$O$_2$; t$^{18}$O$^{12}$C$^{16}$O; ton-year accounting

## 1. Introduction

Forest carbon net ecosystem exchange (NEE) for US-Ha1 employing eddy covariance (EC) for direct in situ $CO_2$ measurement for research purposes has been reported [1–6]. An objective of this study focuses on the requirements for implementing and validating a commercial protocol for net forest carbon storage products suitable for financial market transactions, requirements that have not been elucidated. Existing forest carbon protocols, based on exclusion of direct $CO_2$ measurement, limited forest mensuration, and use of forest growth models [7], are ineffective, resulting in offsets representing ~4% of voluntary carbon trading transactions worldwide in 2019 [8,9], despite ~20+ years of protocol usage [10]. In contrast, GHG trading platforms are expanding rapidly, outpacing the development and implementation of improved direct measurement protocols for forest

$CO_2$, $N_2O$, and $CH_4$ [8,11]. Over 60 carbon pricing initiatives are in place or scheduled for implementation, consisting of 31 emission trading systems and 30 carbon taxes (46 national, 32 subnational jurisdictions) [8], all of which could incorporate directly measured forest carbon product trading. Typically, diverse offsets are differentiated and purchased as voluntary or compliance products priced in 2019 approximately at \$1.62 and \$25 $tCO_2$ (annual averages), on 36.7 million and 8734 million annual totals for $tCO_2$equivalent ($CO_2$e) [12], respectively [9]. Price differentials for carbon offsets, however, are based on similar estimation-based protocols that exclude $CO_2$ precluding equitable and justifiable fundamental prices on forest carbon. Direct measurement and verification of GHG's across forest project sites would fundamentally improve trading and pricing transactions while harmonizing GHG markets across trading platforms [5], another objective of this study. Improved forest carbon protocols will benefit local-to-global forest management projects [13] and diverse landowners including Indigenous People [14] and land subject to continued deforestation of ~4.7 million hectares per year in 2010–2020 [15].

In this study we describe the financialization process for NEE (gC $m^{-2}$ $year^{-1}$) forest products through a retrospective analysis of the longest running forest carbon EC platform in the world, the Harvard Forest, Petersham, MA, USA. Model based forest carbon protocols that exclude $CO_2$ measurement cannot support equivalent transactions or be subject to retrospective analysis and validation employing direct $CO_2$ measurement methods [4]. We analyze the temporal and spatial dimensions of the US-Ha1 Environmental Measurement Site NEE $CO_2$ flux partitioned into Gross Primary Productivity (GPP) and Ecosystem Respiration ($R_{eco}$) including extrapolation of US-Ha1 tower data to the Prospect Hill area (~1500 hectares (ha)), where the US-Ha1 EC tower is located, and to 40,468 ha, based solely on numerical projection, to illustrate potential project revenue with increasing land area. We demonstrate how NEE data relates to financial outcomes, including errors, discount and exit scenarios, fundamental to contemporary commercial carbon trading protocols but as yet not elucidated for entrepreneurs, investors, and landowners for a single long term $CO_2$ flux project. We propose that isoflux for NEE of $t^{13}C^{16}O_2$ and $t^{12}C^{18}O^{16}O$, $CO_2$ isotopocules ($^{ISO}$NEE), be used to partition daytime $R_{eco}$ [16–18] as the basis for new carbon trading offset products that also characterize forest water- and carbon-use efficiency. We discuss the implications of the study for commercialization of NEE and the essential role of $R_{eco}$ in tracking and monetizing the soil carbon assets of forests and related landscapes and limitations of the study.

## 2. Materials and Methods

### 2.1. Eddy Covariance and Net Ecosystem Exchange

Subsequent to calculating NEE with the EC technique, it is usually partitioned into the components of $R_{eco}$ and GPP, according to Equation (1). The partitioning is done using models that predict the rates of $R_{eco}$ during daylight hours from nighttime observations (when GPP is zero) [19,20]:

$$NEE = R_{eco} + GPP \tag{1}$$

In Equation (1), $R_{eco}$ includes the contribution of $CO_2$ from heterotrophic respiration ($R_h$) emitted by soil microbes and fauna, and autotrophic respiration ($R_a$) emitted by vegetation; these components are not partitioned by typical NEE methods [16,17]. NEE is based on established science and provides the commercial integrated carbon quantity of interest, net sequestered carbon or NEE employing widely used algorithms [19] in 600+ publications [21]. It can also be expressed as net ecosystem production (NEP): NEP = −NEE = (−GPP) − $R_{eco}$ [22]. While NEE does not directly probe soil carbon pools, $R_{eco}$ provides a universal method to assess and quantify dynamic soil carbon sequestration, an approach that characterizes flux rather than bulk carbon pools that are regarded as static over time [23,24]. The use of $^{13}CO_2$ isotopic flux partitioning (IFP) [25], particularly attractive for forests (e.g., C3 woody species [26]), further differentiates and quantifies daytime and nighttime respiration providing the basis for new NEE derivative commercial high precision derivative products (e.g., Day $R_{eco}$, Night $R_{eco}$) in contrast to non-isotopic

NEE measurements and model based methods [27]. The need for improved quantification of soil carbon has been identified [28–32].

Fluxes derived from the EC methodology are commonly measured at micrometeorological towers with high-frequency instruments (10 Hz) and averaged over 15, 30, or 60 min periods to isolate turbulence effects from other scalar influences [33]. The towers commonly include other instruments to measure meteorological variables such as air and soil temperatures at different heights and depths ($T_{air}$; $T_{soil}$), incoming global radiation (Rg) or photosynthetically active radiation (PAR), vapor pressure deficit (VPD) or relative humidity (rH). As the measurement sites might be located in remote places in the ambient environment 24 h a day, 365 days a year, it is common to have ~35% of missing data [34] due to system malfunctioning that cannot be repaired in real-time. For instance, storms, snows, high winds, and the insects or animals that live in the forest can damage the instruments, causing a break in data until repairs can be made, often resulting in gaps of days to weeks. Moreover, EC methodology can underestimate NEE over low turbulence conditions [35], thus these points have to be removed. This process may increase the number of missing data points to over 50% [36], but it is necessary to avoid biased data. The friction velocity ($u_*$; Equation (2)) is a measure of turbulence, thus a $u_*$ threshold can be used to classify the data into low and high turbulence periods. Some methodologies to estimate the $u_*$ threshold are described in [35,37].

$$u_* = \left(-\overline{u'w'}\right)^{1/2} \tag{2}$$

To calculate a $CO_2$ budget over a year or any time frame, it is necessary to integrate the flux value over the period, that is, to sum NEE measured at each interval over the whole period. We argue here that the universally accepted relationship (1), established by over 600 publications, is a requirement for the commercialization of forest carbon sequestration, and, further, we argue that the soil carbon component designated as a commercial entity cannot be separated from its above ground counterpart (e.g., forest, agriculture, grazing). The EC methodology integrates soil $CO_2$ efflux ($R_{eco}$) and photosynthesis (GPP) (e.g., (1)). As we explore in this study, EC methods require careful analysis and determination of system uncertainty to be used across project sites and ultimately as a foundation for harmonized NEE carbon commercial products.

Therefore, the gaps in the NEE time series have to be filled. Lookup tables are commonly used to fill the gaps, which consists of creating a table with mean NEE values over several ranges of temperature and radiation jointly and fill missing values with their corresponding mean in the table [34]. For instance, all the missing NEE data points that are between 100–150 W m$^{-2}$ in Rg and 20–21 °C in $T_{air}$ are filled with the mean NEE, calculated with measured data, for that interval. Some variations of this methodology include VPD and different time windows to calculate the means [19]. $CO_2$ budgets estimated with this methodology have a maximum error of around 180 gCO$_2$ m$^{-2}$ year$^{-1}$, depending on the ecosystem type and the quantity of missing data [34].

By signs convention, positive NEE values represent an emission of $CO_2$ from the vegetation to the atmosphere and negative values are $CO_2$ sequestrations. The first occurs mostly at night and are associated with $R_{eco}$ while the latter values are present in day-hours due to GPP being higher than $R_{eco}$.

Partitioning NEE (Equation (1)) into $R_{eco}$ and GPP is useful to better understand how the ecosystem functions and to evaluate hypothetical scenarios, including diagnostic indicators for the sequestration strength of a desired soil carbon commercial product. As the sensor does not distinguish between both fluxes (it only measures NEE), $R_{eco}$ and GPP have to be calculated with available methodology [19,20,32]. For this calculation, the nighttime approach (NT) estimates $R_{eco}$ fitting the Lloyd and Taylor [38] model for respiration (Equation (3)) using only nighttime data, because NEE = $R_{eco}$ at night, and then extrapolating the parameters $R_{ref}$ and $E_0$ found in the regression to calculate daytime $R_{eco}$ ($T_{ref}$ and $T_0$ are fixed). Then, GPP is calculated by difference with Equation (1) [19]. Recent

research suggests that the $R_{eco}$ extrapolation might be biased [32] due to the respiration photo-inhibition of leaves, and it suggests novel methods to partition the flux. However, these methods are under development and most of the references up to date use the NT approach or the daytime approach (DT) that can be found in [20].

$$R_{eco} = R_{ref} \; e^{\frac{E0}{Tref-T0} \; - \; \frac{E0}{Tair-T0}} \tag{3}$$

While all living organisms respire and increase $R_{eco}$, only vegetation contributes to GPP, thus NEE yearly budgets may oscillate between positive and negative values over many years, following changes in the climatic and biological variables. However, data from the longest EC flux measurements tower (Harvard Forest Eddy Covariance Tower) showed that this ecosystem remained as a carbon sink for 15 years [3,6] without any year being a net emission. Therefore, it is important to determine the causes of this pattern to improve the management over similar ecosystems, which may turn to be profitable commercial projects in the international agreements crediting framework.

The Harvard Forest (HF) Environmental Measurement Site tower (42.537755° N, 72.171478° W; US-Ha1) is a component of the Harvard Forest Long Term Ecological Research (LTER) site in Petersham, Massachusetts, USA (Harvard Forest Long Term Ecological Research Site, 2019), and a core site in the AmeriFlux network (US-Ha1) and the National Ecological Observatory Network (NEON) (HARV) (https://www.neonscience.org/field-sites/harv). US-Ha1 is the one of the most intensively studied forests in the world with an elevation of 340 m, an approximate area of ~1500 ha and is classified as Deciduous Broadleaf Forests (DBF; Lands dominated by woody vegetation with a percent cover above 60% and height exceeding 2 m. It consists of broadleaf tree communities with an annual cycle of leaf-on and leaf-off periods). The area surrounding the tower is dominated by red oak (*Quercus rubra*) and red maple (*Acer rubrum*), with scattered stands of Eastern hemlock (*Tsuga canadensis*), white pine (*Pinus strobus*) and red pine (*Pinus resinosa*) with an approximate age of 90–125 years old [3]. The growing season usually starts around mid-May and lasts about 160 days. Canopy height is 20–24 m and the tower height is 31 m. Boston and Hartford are the nearest urban areas, located at 100 km east and 100 km southwest respectively [39]. The climate has a mean annual temperature around 6.5 °C with annual precipitation near 1000 mm, distributed approximately evenly throughout the year. The Harvard Forest trend of net carbon sequestration was first reported in 1993 [1], followed by publications in 2001 [2], 2007 [3], and finally in 2020 [6]. This study considers the NEE record for the EMS tower, although two additional eddy flux towers, the Hemlock and clear-cut towers, have been in operation for shorter periods of time. A large portfolio of soil respiration measurements over 22 years have also been made totaling ~100,000 data points [40]. See [6] for additional details of the measurements and data analyses for the Harvard Forest. The climatic drivers of carbon change, while of relevance and interest to this study, are secondary to the pricing translation of data to carbon markets, representing financial transactions for products as described on offer. In this study periods of net carbon sequestration, according to values of NEE, are considered to be of the highest value to carbon markets, although external factors may also influence carbon prices such as macro and micro-economic trends [41] and regulatory mandates [42]. Biometric and mineral soil carbon analyses are not addressed directly in this study (see [6] for this information).

### 2.2. Data

Carbon dioxide fluxes data [3] were retrieved from the AmeriFlux network (https://ameriflux.lbl.gov/sites/siteinfo/US-Ha1). It is a dataset ("US-Ha1 28-years") with hourly NEE values calculated with the EC methodology, and other meteorological variables like PAR, $T_{air}$ and $T_{soil}$ at different heights and depths or VPD, among others. It started in August 1991 and continues to December 2019. In this work, NEE values outside the range (−60; 40) micromole m$^{-2}$ s$^{-1}$ were labeled as outliers and discarded.

The R package ReddyProc 1.2.1 [36] was used to filter low turbulence periods, to gap-fill the data, and to partition NEE into $R_{eco}$ and GPP. It requires some meteorological variables as input data, besides the fluxes, which are Rg, $T_{air}$, $T_{soil}$, and VPD or rH. As in the dataset are many possibilities to choose for each variable, the ones with less missing data were used, which were: above canopy $T_{air}$ (TA_PI_F_1_1_1) and rH (RH_1_1_1), $T_{soil}$ measured at the lowest depth (TS_PI_1), and PAR (PPFD_IN_PI_F_1_1_1) divided by 0.47 as Rg [43,44]. Carbon flux (FC) and carbon storage (SC) were summed to complete NEE.

To filter low turbulence periods, the software calculated yearly $u_*$ thresholds following [35] using the 50 percentile criterion and it discarded the points with $u_*$ below that bound. Then, it created a Look-Up Table with the remaining data points, as described in [19], to fill the missing NEE values. NT methodology was selected over DT or others to partition the flux because, up to date, there are more publications to compare the results.

After processing all the data, weekly, monthly, yearly, and multi-yearly NEE, $R_{eco}$, and GPP budgets were calculated to describe inter-annual and intra-annual variability, as well as their financial extrapolations. An online platform displaying daily US-Ha1 data (previous day) and cumulative values can be found here: www.pemcarbon.com/ecaas/. We note that extrapolation of US-Ha1 flux tower data to larger areas is purely for illustration. Upscaling the US-Ha1 results to surrounding areas requires additional analysis including remotely sensed data and scale-aware models and are not addressed in this study.

### 2.3. Soil Carbon Data

Data for soil respiration and for bulk soil organic content used in this study were provided by [6,40]. The dataset employed represents a recompilation of many datasets measured across twenty-three studies listed in [40]. Four methods were used to measure $R_s$: (1) soda-lime systems where pellets were left beneath a closed chamber for 24 h to absorb $CO_2$ emitted from the soil; (2) static chamber systems where a chamber was placed on a collar inserted into the soil and headspace air samples were taken at fixed intervals over 15 to 30 min and subsequently analyzed with an infrared gas analyzer (IRGA) or a gas chromatograph; (3) dynamic chamber systems in which a chamber was placed on each collar, chamber air was circulated to and from a portable IRGA system, and the rate of increase in $CO_2$ concentration was measured in situ for a period of five minutes; (4) and automated chamber systems, in which a datalogger-controlled system closed one chamber at a time and circulated the headspace air through an IRGA [6,40].

### 2.4. Carbon Isotopocules and $^{iso}$NEE

The isotopocule $^{13}C^{16}O_2$, and $^{18}O^{12}C^{16}O$ data were measured between April and October from 2011 to 2013 at the Harvard Forest and quality-controlled as the dataset "hf209–10" [17] downloaded from https://harvardforest1.fas.harvard.edu/exist/apps/datasets/showData.html?id=209 where a detailed explanation of the measurement system can be found. It has $^{13}CO_2$ isofluxes and $^{12}CO_2$ fluxes obtained at irregular intervals of around 40 min with an approximate 50% of missing data that were used to calculate $^{13}C$ yearly budgets. The isotopocules molar mixing ratios were measured by a quantum cascade laser spectrometer, which runs in a temperature-controlled enclosure. Then, they were calibrated on a 40-min interval by linear interpolation between two cylinders: a high span gas cylinder containing roughly 450 ppm $CO_2$ in air, and a low span cylinder containing roughly 350 ppm $CO_2$ in air. The total $CO_2$ molar mixing ratio (to dry air) and the isotope ratios were calculated from the individual isotopocule molar mixing ratios measured by the spectrometer. Eddy and storage fluxes for each isotopocule and for total $CO_2$, plus the eddy and storage isofluxes, were calculated by EC using wind data from the sonic anemometer that is operated as part of the existing long-term eddy flux measurement system. The results were priced in \$50 t$C^{13}O_2$ and \$150 t$C^{13}O_2$ pricing scenarios (see pricing information below) with extrapolations to 300 ha and 40,468 ha, based solely on numerical projection to illustrate potential project revenue with increasing land area. To explore the relationship between $^{13}C$ and NEE from both datasets ("US-Ha1 28-years" and "hf209-10"), hourly

$^{13}CO_2$ means were used. All regressions and statistics were calculated using Python 3.7 and SciPy 1.4.1 library (https://docs.scipy.org/doc/).

Isotopic measurements for $^{13}C$ are expressed as ratios in per mil (‰) in standard notation relative to Vienna Pee Dee Belemnite (VPDB) standard [45]. The purpose of employing metric tons of $^{13}C^{16}O_2$, as defined for $t^{12}CO_2$, is to further partition and quantify soil $CO_2$ respiration since $R_{eco}$ as reported for NEE represents both soils based heterotrophic ($R_h$) and vegetation based autotrophic ($R_a$) carbon efflux measured during the night when GPP = 0. In contrast, isotopic flux partitioning (IFP) [16–18,46] utilizes the inherent discrimination against the heavier carbon isotope ($^{13}C$) resulting in fractionation at the leaf level by the Rubisco enzyme during photosynthesis between atmospheric and plant matter of up to ~29 per mil (‰) for plants with the C3 photosynthetic mode [26]. The isotopic imprint of reduced or isotopically lighter $^{13}C$ composition remains locked in plant matter until it is transferred to the atmosphere as $^{13}CO_2$ during decomposition by microbes (e.g., $R_a$) differentiating respired from isotopically heavy tropospheric $CO_2$. The fractionation occurs instantaneously at the leaf level imprinting organic matter as substrate for respiration and isotopic measurement. The actual $\delta^{13}C$ of vegetation and subsequent release of soil $CO_2$ via respiration for a given sample will vary across the temporal changes in atmospheric $\delta^{13}C$ and rainfall experienced during growth among other factors [26]. For example, tree ring $\delta^{13}C$ for two tree species of the Harvard Forest, *Quercus rubra* and Tsuga canadensis, ranged from ~$-22.3$ to $-24.7$ ‰ across seven annual rings (1997–2003) [47] while mean annual precipitation and canopy effects of ~$-28$ to $-33$ ‰ $\delta^{13}C$ have been reported for tropical forests [48]. The $\Delta-\delta^{13}C$ between the atmosphere and plant matter diminishes for plants with the dicarboxylic acid (C4) pathway plants (e.g., grasses, crops) to ~2‰ [26,49]). Eddy covariance can be employed to determine fluxes for carbon isotopocules (e.g., $^{13}C^{16}O_2$, and $^{18}O^{12}C^{16}O$, or 636 and 826, respectively, in HITRAN notation [50], with natural abundances of 0.011057 and 0.003974, respectively) in much the same way as for bulk $CO_2$ (e.g., $^{12}CO_2$, $^{13}CO_2$) but with a high precision isotopic analyzer (e.g., ~4 Hz response and time averaging over the EC integration period of 30 to 60 min). In this study, 636 and 826 are analyzed: commercial analyzers are available for 726. The 826 isotopocule is of relevance to intrinsic ecosystem water and carbon use efficiency [16]. The carbon isotope ratio $^{13}C/^{12}C$ is approximately equal to the isotopocule ratio $^{13}CO_2/^{12}CO_2$; isoflux of $^{13}C^{16}O_2$ is proportional and highly correlated with its $^{12}CO_2$ counterpart. Isofluxes of 626 and 826 (‰ µmol m$^{-2}$ s$^{-1}$) result in net isoflux $I_N^{13} = \sigma_N^{13} F_N$, [17] or, for carbon trading, expressed in metric tons as $^{ISO}$NEE $t^{13}C^{16}O_2$ and $^{ISO}$NEE $t^{18}O^{12}C^{16}O$. Commercial, off-the-shelf carbon isotopic analyzers and typical EC components are available as described in [16–18,25].

### 2.5. Carbon Pricing

We have selected carbon pricing to illustrate the financial value of NEE had it been sold at \$10 and at \$30 $tCO_2$. A price of \$10 reflects the initial price floor for CARB forest carbon offsets [51]. A price of \$30 reflects the lower end of estimated carbon pricing to cover the social cost of carbon [52]. Isotopocules of $CO_2$ have not been proposed or priced as carbon financial instruments, however, NEE isoflux for $^{13}C^{16}O_2$ is expected to provide partitioning of $R_{eco}$ into fluxes for $R_a$ and $R_h$, reducing the uncertainty of $CO_2$ efflux, and thus should command a higher price than undifferentiated $R_{eco}$. We use the arbitrary pricing of \$50 and \$150 to illustrate the potential financial value of carbon isotopocules.

### 2.6. Ton-Year Accounting

In the Kyoto Protocol [53], emission avoidances are considered to be permanent, whereas $CO_2$ captured and stored in trees or wood products may return to the atmosphere due to fire or decomposition, creating a need for an equivalence system to quantify both activities contributions to the climate change mitigation [54]. Up to date, there are at least 15 different methodologies to account for this, with no agreement of the most appropriate [55]. One of these is the ton-year Moura-Costa [56] accounting method, selected for this study because it is the simplest and easiest to understand for a nontechnical audience, which

makes it a more attractive option to this kind of investors. It has an Equivalence Time (Te) and it is estimated according to Equation (4) (M-C), but with Te calculated following the revised Bern Model (Equation (5)), which is 46 years [54]. Therefore, if one $tCO_2e$ credit is equivalent to delay an emission for 100 years, M-C assigns 1/46 $tCO_2e$ credits to a $CO_2$ ton sequestered for one year, 55/46 $tCO_2e$ credits to a $CO_2$ ton stored for 55 years, etc.

$$tCO_2e = \frac{tCO_2 - tCO_{2\ baseline}}{Te} \tag{4}$$

$$[CO_2(year)] = 0.18 + 0.14\, e^{\frac{-year}{421}} + 0.19\, e^{\frac{-year}{71}} + 0.24\, e^{\frac{-year}{21}} + 0.26\, e^{\frac{-year}{3.5}} \tag{5}$$

The M-C ton-year accounting method, supposing $tCO_{2\ baseline} = 0$, was used to explore 5, 10, 15 and 20 years exit options profitability, along with Net Present Value calculations (NPV; Equation (6).) of a hypothetical project that started in 1992 and finished at any year, discounted at 0%, 1%, 3%, 5%, 10%, 15% discount rates. These results were compared with the same projects but compensated with the California Air Resources Board (CARB) protocol. Also, they were extrapolated to a hypothetical 40,468 ha (100 k acres) land and the Prospect Hill area (300 ha; 741 acres), while different carbon price scenarios were applied ($10 or $30 a $tCO_2e$) to analyze the financial value of both lands.

$$NPV = \sum_{year} \frac{Cashflow_{year}}{(1 + interest)^{year}} \tag{6}$$

### 2.7. Study Limitations

We note that extrapolation of US-Ha1 flux tower data (~10 ha) to landscape scales (e.g., 40,468 ha) is based solely on simple numerical projection to illustrate potential project revenue with increasing land area, not an interpretation of ecological net ecosystem exchange for the region. It is widely acknowledged that one of the primary limitations of eddy covariance based NEE is the uncertainty of upscaling limited footprints for individual EC towers to surrounding ecosystems [57]. The financial scope of NEE revenue illustrated by projection is the counterpart to scientific data bridging the gap between science and commerce and is justifiable for this purpose. Eddy covariance up-scaling including the use of EC networks in the context of a direct measurement forest carbon protocol have been described previously [5,58]. Up-scaling of the US-Ha1 NEE results from the tower footprint to surrounding areas would require additional EC platforms to cover gaps, ecological analysis, remotely sensed data and the use of scale-aware models that are beyond the scope of this study [59–66]. This study is also subject to limitations and uncertainties of the NEE methodology itself including accounting for periods of advective and low-frequency flows of $CO_2$ that are difficult to capture leading potentially to underestimation of fluxes; the results used in this study have been corrected for these conditions as described (Methods, 2.1 Data) [6]. Lack of EC replication and a persistent inability to close the surface energy budget have been noted [67]. While these limitations have been addressed to the extent possible for the US-Ha1 data [6], details of standardization of QA/QC thresholds [35,68], and upscaling using remote sensing data [57,61,69,70] are outside of the scope of this study. Due to the short-term EC results for two additional towers reported for the US-Ha1 research area, the Hemlock tower (11 years) and clear-cut tower (6 years) [6], these results were omitted in the 28-year long term analysis presented.

### 3. Results

*3.1. The Full Harvard Forest NEE Record*

The full US-Ha1 NEE records are shown in Figure 1A for NEE, $R_{eco}$, and GPP hourly values after filtering and gap-filling the full US-Ha1 record; biomass removal was negligible for the intervals analyzed. Figure 1B aggregates these variables as yearly sums identifying three main intervals with different NEE behavior. Up to 2003, the sequestration was approximately constant, with a yearly NEE mean ($\bar{x}$) of $-703$ gCO$_2$ m$^{-2}$ year$^{-1}$ and

an associated standard deviation (SD) of $\pm$ 239 gCO$_2$ m$^{-2}$ year$^{-1}$. The next six years (2004–2009) exhibited higher net sequestration with a peak in 2008, when the NEE $\overline{x}$ and SD increased to $-1658 \pm 414$ gCO$_2$ m$^{-2}$ year$^{-1}$, respectively. For the period 1992–2008, the $\overline{x}$, SD were: $-1007$ and 563 gCO$_2$ m$^{-2}$year$^{-1}$, respectively, with a trend of $-86.8$ gCO$_2$ m$^{-2}$year$^{-1}$, R$^2$: 0.61. There was an abrupt decline in NEE in 2009–2011 to a value of $-4.21$ gCO$_2$ m$^{-2}$ year$^{-1}$. Between 2012 and 2019, NEE became more unstable ($\overline{x} = -1093$ gCO$_2$ m$^{-2}$ year$^{-1}$ $\pm$ 472 gCO$_2$ m$^{-2}$ year$^{-1}$) than in previous periods but remained a net carbon sink. Over the whole term (1992–2019), the yearly mean carbon sequestration was $-978$ gCO$_2$ $\pm$ 553 gCO$_2$ m$^{-2}$ year$^{-1}$ [6]. NEE reached its minimum in 2010 ($-4.21$ gCO$_2$ m$^{-2}$ year$^{-1}$) and its maximum in 2008 ($-2199$ gCO$_2$ m$^{-2}$ year$^{-1}$). Carbon uptake was distributed unevenly during the year, with sequestration during the forest growing season and emissions in the remaining weeks. NEE weekly sums for winter and summer accumulates carbon annually resulting in 27,101 gCO$_2$ m$^{-2}$ for the 1991 to 2019 period. Figure 1C shows soil respiration (R$_s$) divided by Ecosystem Respiration (R$_e$) yearly values (Brown) and total soil carbon measured in the top 15 cm of the mineral soil of the hardwood- (purple) and conifer- (grey) dominated plots in 1992 and 2013 from [6] to illustrate the coupling between diverse biological and financial indices for the US-Ha1 record. In Figure 1D we extrapolate the US-Ha1 data to a larger area (40,468 ha or 100,000 acres), based solely on numerical projection to illustrate potential project revenue with increasing land area, to pricing at the initial floor price for CARB of \$10 and above the suggested social price of carbon of \$25. The period 2009–2012 is highlighted with a grey bar to emphasize the change in ecosystem function across the indicators, for example, increased R$_{eco}$ and decrease in R$_s$/R$_e$ shown in Figure 1B,C, respectively, and propagation of carbon loss to financial loss shown in Figure 1D.

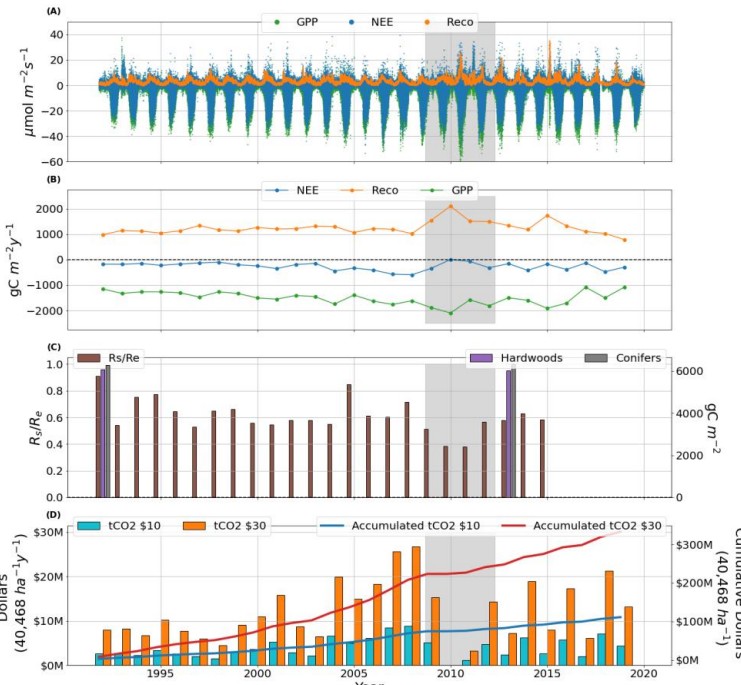

**Figure 1.** (**A**) Hourly NEE (blue), GPP (green), and R$_{eco}$ (orange) for the period 1992 to 2019. (**B**) Same as (**A**) shown as yearly sums. (**C**) Soil respiration (R$_s$) divided by Ecosystem Respiration (R$_e$) yearly values (Brown) and total soil carbon measured in the top 15 cm of the mineral soil of the hardwood- (purple) and conifer- (grey) dominated plots from [6]. (**D**) CO$_2$ yearly sequestration value in U.S dollars (\$) assuming a ton of CO$_2$ were worth \$10 (light blue bars) or \$30 (orange bars), and their accumulated values (blue and red lines respectively) extrapolated to 40,468 ha (100,000 acres), based solely on numerical projection to illustrate potential project revenue with increasing land area, assuming there were no exits during the interval 1992–2019. Further details of this calculation are available in Appendix A. The period 2009–2012 is highlighted with a grey shade to illustrate status of variables resulting in large financial loss for the interval.

Referring to Figure 1D, pre-tax, gross income from the Harvard Forest for the total period of data would be over $109 million, or over $329 million for a price of $30 tCO$_2$. Additional data for the daily and seasonal NEE, GPP and $R_{eco}$ behavior can be found in Appendix A (Figure A1, hourly and yearly NEE; Figure A2, Annual and weekly histograms for NEE, $R_{eco}$ and GPP; Figure A3, annual and monthly $R_{eco}$ versus GPP, and Table A1 for yearly statistics) and online: https://pemcarbon.com/ecaas/.

### 3.2. Box Plots of Project Time Interval and Area Extrapolations

Time and area comparisons for NEE are provided in Figure 2 shown as box plots for the full record of no exit, 5 and 20 year exits, and for the CO$_2$ isotopocules for $^{ISO}$NEE t$^{13}$C$^{16}$O$_2$ and $^{ISO}$NEE t$^{12}$C$^{18}$O$^{16}$O isofluxes. Figure 2A shows that the no exit 28-year project was more volatile considering values for first and third quartiles and outliers, than for a 20-year exit (median values: $79.5 and $21.5 ha$^{-1}$ year$^{-1}$ respectively, in a $10 tCO$_2$e pricing scenario), and that carbon isotopocule isoflux products would command higher per-ton price levels than non-isotopic counterparts, in line with enhanced partition values for $R_{eco}$ and R$_a$. Figure 2B.

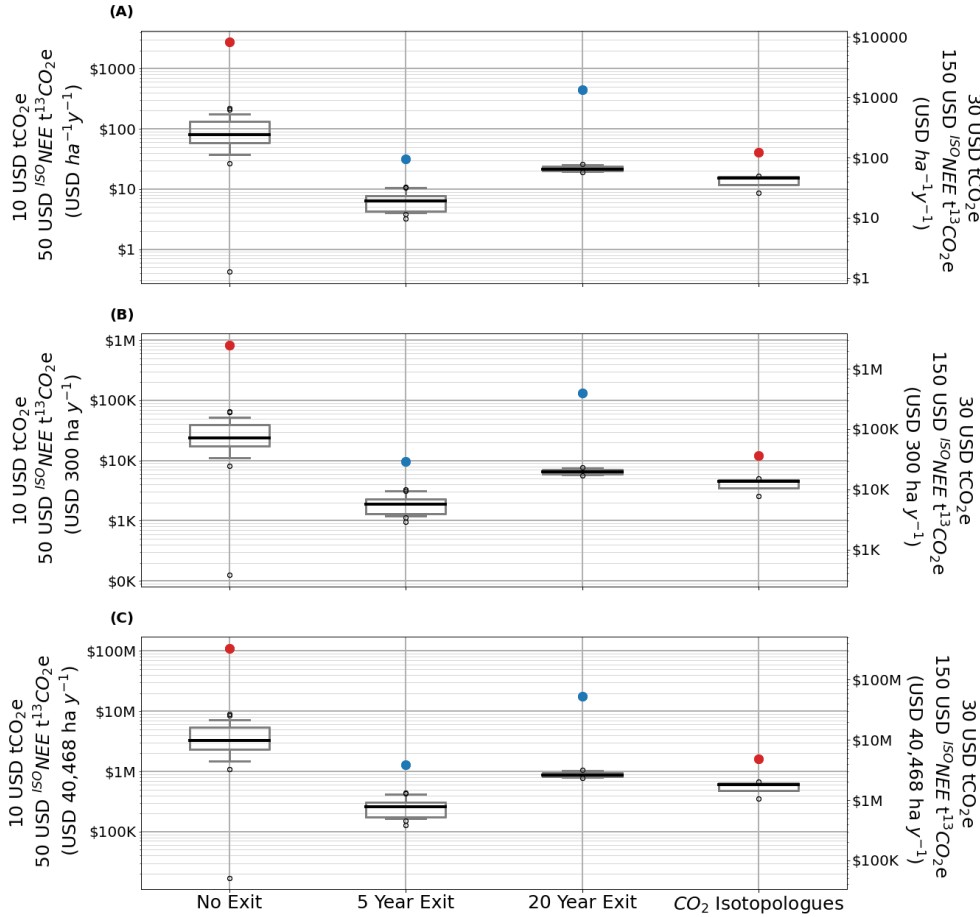

**Figure 2.** Yearly financial CO$_2$ and $^{13}$CO$_2$ (or $^{12}$C$^{18}$O$^{16}$O) box plots with median (black line), first and third quartiles (box), 5 and 95 percentiles (whiskers) and outliers (circles) per ha (**A**), extrapolated to Prospect Hill area (300 ha; (**B**)) and extrapolated to 40,468 ha (100,000 acres; (**C**)), based solely on numerical projection to illustrate potential project revenue with increasing land area, in logarithmic scale. The red points are accumulated values over the whole period; blue points are mean accumulated values at the end of exited projects (5 or 20 years, respectively). Left and right *y*-axis show different pricing scenarios (left: a ton of CO$_2$ is worth $10 and a ton of $^{13}$CO$_2$ (or $^{12}$C$^{18}$O$^{16}$O) $50; right: a ton of CO$_2$ is worth $30 and a ton of $^{13}$CO$_2$ (or $^{12}$C$^{18}$O$^{16}$O) $150).

Figure 2C shows the potential magnitude of pre-tax gross revenue for 300 and 40,468 ha, based solely on numerical projection to illustrate potential project revenue with increasing land area, $813 thousand and ~$109 million, respectively, priced at $10 tCO$_2$. The financial values presented in Figure 2A–C and Figure 1D assumes that carbon storage is permanent, an assumption that does not apply to forests.

### 3.3. Ton-Year Accounting Applied to the Harvard Forest Record

Referring to Figure 3A, application of this approach to five-year US-Ha1 projects would have earned around $30 or $90 ha$^{-1}$ with a $10 or $30 tCO$_2$ price scenario, respectively, while longer projects (e.g., 10, 15, 20 years) would have returned multiples of the five-year earnings because they sequestered more carbon in that period. Note that the CARB exit represents a zero return for reforestation (Refo) and avoided emissions (AvCon) projects and a negative return for improved forest management (IFM) CARB projects. Figure 2B shows that yearly mean returns (i.e., the total return divided by the length of the project) would have also been higher for longer projects, in this case, because they stored carbon for more time, which is better compensated in ton-year accounting methods. The CARB protocol would have resulted in 100% payback of the project offset value by landowners to CARB and or penalty to the landowners and investors if the project ended before 100 years according to project type as in Figure 3A.

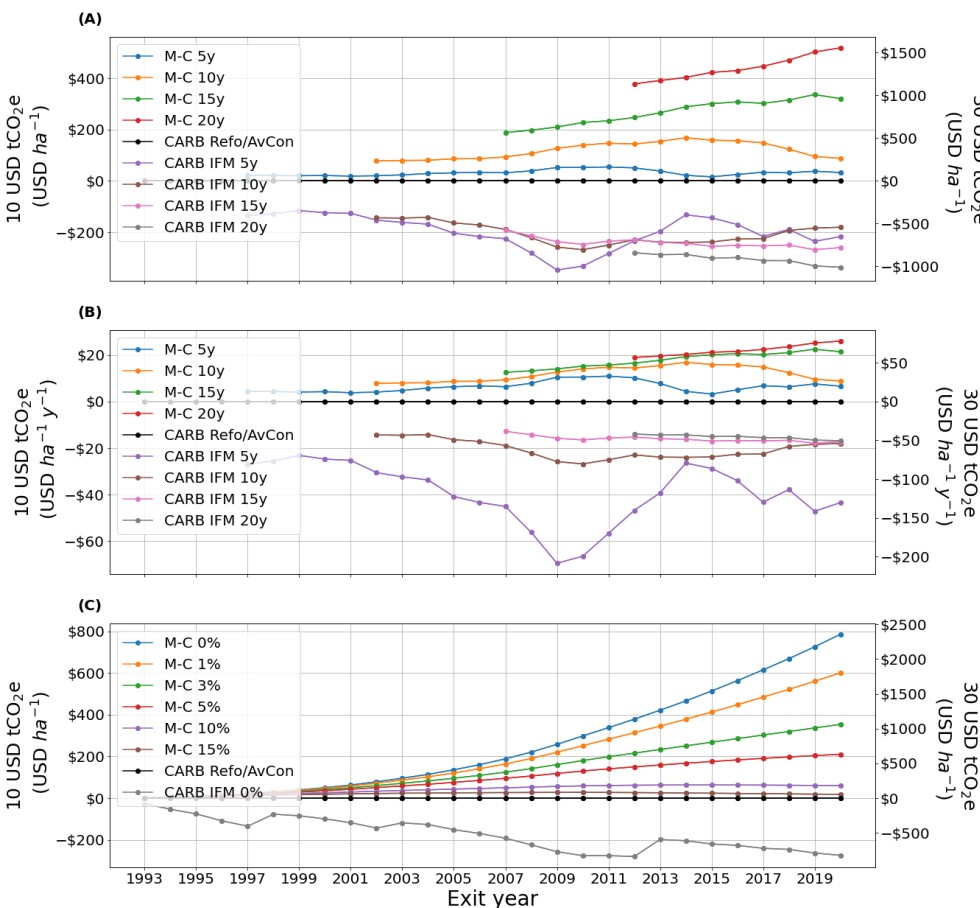

**Figure 3.** (**A**) Ton-year accounting return per ha for projects that stored carbon for 5 (blue), 10 (orange), 15 (green), and 20 (red) years, using the Moura-Costa approach, and CARB protocol returns for reforestation and avoided conversion projects (Refo/AvCon; black), for improved forests management (IFM) in 5 (purple), 10 (brown), 15 (pink), and 20 (grey) years. (**B**) Same as (**A**) but showing yearly mean returns. (**C**) Returns per ha using the same carbon pricing scenarios as (**A**) for projects that started in 1992 and finished in one exit year, discounted at 0% (blue), 1% (orange), 3% (green), 5% (red), 10% (purple), and 15% (brown) calculated with the Moura-Costa approach and with both CARB protocols without discounting (black and grey). In this case, there are 28 projects for each discount rate with different lengths.

### 3.4. CO$_2$ Isotopocules as Tradable Forest Carbon Products

The results for $^{ISO}$NEE$^{13}$C$^{16}$O$_2$, and $^{ISO}$NEE $^{18}$O$^{12}$C$^{16}$O are presented as new carbon trading products that are monetized in a manner similar to that for $^{12}$CO$_2$ NEE flux. We used the relationships shown in Figure 4A–C, to partition and calculate NEE isoflux as t$^{13}$C$^{16}$O$_2$ and t$^{18}$O$^{12}$C$^{16}$O (see also Figure 2A–C) boxplots and data summary of Table A1). Figure 4A establishes the linear and correlated relationship between the isotopic composition of $^{13}$CO$_2$ flux, of opposite sign, and $^{12}$CO$_2$ flux expected from coupled fractionations shared by leaf and ecosystem CO$_2$ dynamics, as cited in Methods, 2.4. Figure 4B shows $^{13}$C$^{16}$O$_2$ data aggregated and plotted against NEE from the "US-Ha1 28-years" dataset, to establish the basis for the coupled annual record and calculations. Likewise, Figure 4C illustrates the relationship between the non-gap filled $^{18}$O$^{12}$C$^{16}$O data plotted against NEE $^{12}$C$^{16}$O$_2$ non-gap filled data to establish the basis for the annual record and calculations; the data sets available did not provide gap-filled data for this isotopic series. The results identify carbon and oxygen isotopic species as singular or mixed isotopic masses that are quantified and financialized according to isotopic mass across the project temporal and spatial domains. The results (Figure 2A–C) suggest market value for $^{ISO}$NEE as multiples of pricing for non-isotopic counterparts.

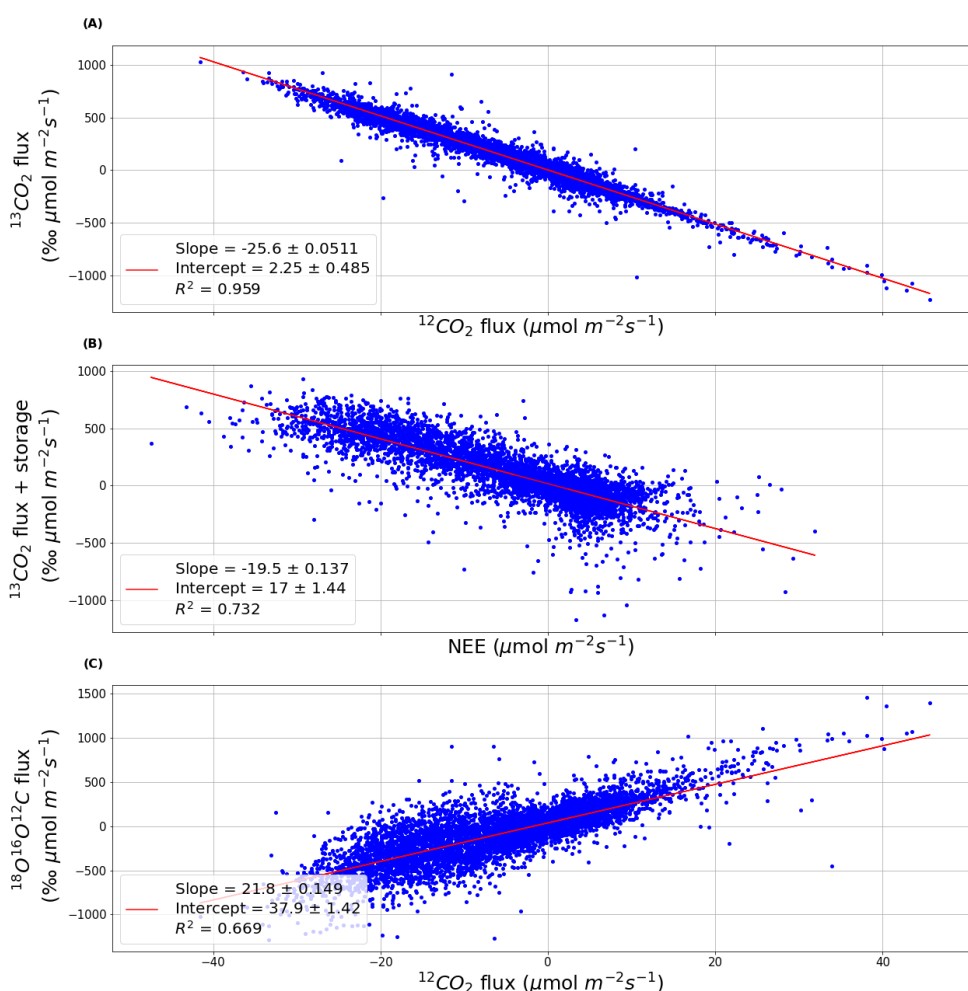

**Figure 4.** CO$_2$ isotopocule data, (**A**) shows the non-gap filled $^{13}$C$^{16}$O$^2$ data plotted against NEE determined using $^{12}$C$^{16}$O$_2$ data ("hf209-10" dataset) between April and October from 2011 to 2013; (**B**) shows $^{13}$C$^{16}$O$_2$ data aggregated to be plotted against NEE from the "US-Ha1 28-years" dataset; (**C**) shows the non-gap filled $^{18}$O$^{12}$C$^{16}$O data plotted against NEE $^{12}$C$^{16}$O$_2$ non-gap filled data.

## 4. Discussion

A universal scientifically accepted methodology based on direct measurement of $CO_2$ flux (EC) and net carbon sequestration (NEE), applied to the US-Ha1 28-year record, establishes the foundation for verifiable commercial carbon products. Relying upon the NEE methodology, forest carbon protocols that exclude $CO_2$ data for $R_{eco}$ would result in a project error (1992 to 2019) of ~466% given that $R_{eco}$ represents ~45% of NEE and cannot be excluded from a full net forest carbon accounting for financial products and trading [4]. As a result, forest carbon model based determinations of sequestered carbon are likely more uncertain as proxy data cannot be validated without independent direct measurement [5]. The level of uncertainty for model based commercial products results in discount pricing observed, for example, in 2019, by forest voluntary carbon offsets priced typically < \$1.67 USD for the highest volume of offsets under the Verified Carbon Standard [9]. Direct measurement of NEE $CO_2$, as described in this study, provides a harmonized and equivalent pricing basis for voluntary and compliance pricing [5].

The lessons learned from the long-term retrospective US-Ha1 carbon sequestration record include: (1) financialization of scientific NEE data is readily achievable using existing instrumentation, reporting and transaction mechanisms linking landowners with buyers of forest carbon offsets and expanding carbon markets, (2) considerable flexibility across spatial domains and temporal intervals, discount rates and feasible exit terms can be assessed by buyers and sellers of forest carbon offsets to inform decision making, (3) the permanence of net sequestered forest carbon while shown to be viable for ~100+ years, as for the US-Ha1, is biologically labile to increases in $R_{eco}$ relative to GPP (e.g., 2009–2011), and to external factors including changes in rainfall, surface temperature, extreme events (e.g., El Nino, La Nina) [71], and length of growing season [6] and must be monitored to verify and adjust monetization strategies, (4) $R_{eco}$ lays the foundation for integrated dynamic soil carbon sequestration commercial products ($gCO_2$ $m^{-2}year^{-1}$), in contrast to reliance on bulk soil carbon pool ($gC$ $m^{-2}$) approaches, and, (5) isotopic flux of $CO_2$ as $^{ISO}$NEE $t^{13}C^{16}O_2$, and $t^{18}O^{12}C^{16}O$, further partitions NEE as innovative and derivative forest soil and forest carbon commercial products. Taken together, the use of EC NEE and $^{ISO}$NEE harmonizes carbon sequestration applications across diverse projects (e.g., reforestation, agriculture, grazing, afforestation), and policy platforms (e.g., Paris Agreement [72]; Reducing Emissions from Deforestation and Forest Degradation, or UN-REDD Program catalyzing carbon markets and nature based climate change solutions [73] across contemporary and long-term project time scales.

The NEE commercial forest carbon offsets may also play a role in catalyzing efforts to reverse deforestation against persistent forest loss worldwide of ~4.7 million ha per year over the last decade [15]. Reversing deforestation and protecting conserved forests will be a massive undertaking. For example, in 2019 the value of the voluntary market for forestry and land use projects was ~\$159.1 million USD representing 36.7 million $tCO_2$ [9] compared to the compliance market of ~\$175 billion USD and ~6777 million $tCO_2$ [8]. The potential for forest restoration and conservation projects (e.g., 10 k to 50 k ha project areas) covering 0.92 billion ha could yield an estimated \$345 billion in revenue with verified NEE of 2.76 billion $tCO_2$ for long-term projects. Moreover, short term projects exiting at 5, 10, and 15 years, according to ton-year accounting, yield a value of \$60 billion. The number of projects and areas required (e.g., 10,000–50,000 ha) to achieve the NEE ($tCO_2e$ 2.76 billion) in this hypothetical scenario is 55,200 compared to ~67 CDM forest projects recorded since 2006 [74] (see Appendix A, Table A2 for additional details); in contrast there are ~1.5 million farms in the US alone [75]. The forgoing comparison is a stark reminder of the scale of action needed to reverse deforestation. Impediments to expansion of reforestation projects result, in part, from the high cost and invalidation risk to landowners (e.g., exclusion of $CO_2$ measurement) typical of existing forest carbon protocols [76]. A comparison of features and functions of existing protocols and eddy covariance NEE methods has been reported [5] emphasizing the need for scientific protocol criteria that uncouples pricing constraints from

regulatory and legislative mandates across offset types, equalizing forest carbon project opportunities and earning potential [5] for all landowners including Indigenous People.

We emphasize that existing forest carbon protocols require that each $tCO_2$ captured remains fixed in the trees or soil for at least 100 years fulfilling the arbitrary permanence requirement of non-$CO_2$ based protocols such as the CARB and Climate Action Reserve (CAR) [4,77,78]. According to CARB-CAR protocols, landowners who exit the project prior to 100 years must repay the value of all issued offsets in the cases of reforestation and avoided emissions, and pay an additional penalty in the case of improved forest management [7], making exits punitive and infeasible. In contrast, we show here that accounting methods can accommodate realistic temporary carbon capture and storage without arbitrary exit terms [7], delaying total deforestation and the effects of climate change and providing time intervals to take other actions, for example, reforesting ~0.9 billion ha of degraded land [79]. The Moura-Costa ton-year accounting method [56] combined with the revised Bern Model for decay of atmospheric $CO_2$ concentration over time [54] provides viable exit options for shorter projects, an approach endorsed by the IPCC [80]. Furthermore, variance in the no exit case for US-Ha1 is higher than in exited projects (Figure 2 and Table A1 from the Supplemental) emphasizing the variable nature of annual carbon sequestration even over long intervals [81] and the need for direct measurement to validate claims of annual net $CO_2$ emission reduction for carbon markets by ecosystems, and $R_{eco}$ as represented by isofluxes. This approach incentivizes many of the potential short projects that are outside any existing net forest carbon protocol because the 100-year required project term is a high barrier to entry for landowners reflected in CARB-CAR projects representing less than ~0.2% of available US land [4,77]. Moreover, the implementation of AB398 imposes new changes in offset pricing including an arbitrary price ceiling of $65 per allowance, and requires at least 50% of projects to directly benefit the State of California implying a carve-out for projects that must originate in the state [82]. The consequences of AB398 are not known but suggests a retraction of the CARB-CAR forest carbon protocol from areas outside of California. Clearly, an alternative protocol that catalyzes forest conservation and restoration that is uncoupled from regulatory legislation is needed, such as that presented in this study.

The US-Ha1 analysis suggests that short- and long-term forest conservation (e.g., minimal timber removal) is viable and accretive to the forest regeneration process. From a commercial perspective, selection of the optimal project length (Figure 3C) depends on the discount rate that the investor uses to account for the time value of money. While low discount rates (below 5%) incentivize longer projects, making their returns rise exponentially as shown in Figure 3C, a rate above 10% would make these decay after a few years. For example, a project that started in 1992 and finished in 2020 (28-years) would have returned less money than one that finished in 2002 (10 years), when discounted at 15%. In practice this means that landowners and investors who value the present more than the future would select shorter projects (below 10 years) compared to a retirement fund with an investment outlook of decades. Referring to Figure 2A–C, 5- and 20-year project exits would have exhibited less yearly physical and financial carbon variance (and SD) than no exit investments. Exits at 5 and 20 years (Figure 2C) (40,468 ha, $10 $tCO_2$) are characterized by $\bar{x}$ of ~$256,000 and $891,000, and SD of ± $93,000 and ± $98,000, respectively, compared to the no exit 28-year project with $\bar{x}$ of ~$3.9 million and SD of ± $2.2 million. Thus, extending a project time frame does not necessarily diminish its financial volatility, because the accounting method would have assigned more credits to longer projects that stored carbon for more years, increasing its return and variance, as shown in Figure 2, whereas the yearly variance would have been higher for a 20-year project than for a five-year project. Investors and project landowners may also consider that if the mean $CO_2$ NEE value was a good estimation of a project expected income, then to earn one dollar at the end of the project an investor would buy 0.16 ha or 0.05 ha (5- or 20-years projects respectively; $10 $tCO_2$). For these project parcels (e.g., 0.16 or 0.05 ha) they would expect to earn $[0.65;1.35] for five-year projects and $[0.89;1.11] for 20-year projects (66% confidence), respectively.

The retrospective US-Ha1 NEE data implies that high annual variance, while unpredictable, can be managed by investment objectives such as time horizon, desired return on investment and reliance on standardized direct measurement methods. Although we have only analyzed the US-Ha1 EMS flux tower results in this study, a portfolio of diverse forest projects (e.g., location, ecosystem type, management objective, and vulnerability to climate change) could buffer volatility reducing the impact of the US-Ha1 NEE decline experienced in 2010 [5], an established benefit of portfolio diversification [83]. Likewise, additional US-Ha1 flux tower experiments documenting decline of NEE for a stand of hemlock forest (HEM) due to insect damage (e.g., hemlock woolly adelgid) and for a clear-cut experiment (CC) demonstrate that networks of EC platforms and NEE measurements can capture and integrate multiple changing NEE forest landscapes [6]. The HEM site transitioned from a net sink (~450 gC $m^{-2}$year$^{-1}$) to source over the period studied (2005–2012). The CC experimental site initial timber harvest (2009) [84] was a large net carbon source but transitioned to an annual net carbon sink after the fifth year of disturbance regaining ~66% of the carbon lost since harvest [6]. The detection of forest project carbon source(sink) transitions is considered a key capability for a forest carbon trading protocol, a well-documented feature of EC NEE, but an insuperable omission for estimation-based protocols and analysis of bulk soil carbon pools.

In the context of establishing an approach relying on GPP and $R_{eco}$ with emphasis on commercial products, multiple measurement methods and innovative approaches, are required to both understand carbon dynamics and transact reliable global forest carbon and soil carbon sequestration markets. Monetary calculations for the US-Ha1 28-year NEE record analyzed and described in this study did not include results for soil carbon flux measurement, summarized in Figure 1C. However, the US-Ha1 coupled EC and soil respiration dynamics record establish an unprecedented experimental biosphere for soil model development and validation. Over 100,000 individual measurements of soil respiration across 23 experimental studies, five different forest types and spanning more than 25 years were conducted with cross-cutting footprints for the US-Ha1 EMS tower data analyzed herein. For example, the relationship between soil $CO_2$ efflux ($R_s$) determined by individual chamber measurements, and total ecosystem respiration, $R_e$, for the US-Ha1 NEE EC record analyzed in this study, covering shared tower footprints, shows that $R_s/R_e$ varied from 0.49 to 0.88 (Figure 1C). The large variation emphasizes the complexity of the biospheric system and seasonal carbon dynamics that challenge model development and monitoring applications. Figure 1C shows that soil chamber measurements ($R_s$) and EC measurements ($R_e$) capture the perturbation of the 2010 US-Ha1 environment resulting in a severe loss of forest carbon asset value with an uneven recovery spanning the period up to 2019. In forest carbon trading practice, a landowner employing EC may also choose to rely upon repeat soil chamber measurements as feasible and cost-effective to improve understanding of the project soil carbon and financial dynamics.

In addition to soil respiration chambers, soil organic carbon measurements were made across the US-Ha1 site footprint [6,40] to characterize soil carbon pools. Results for soil carbon pools revealed negligible change in net accrual of soil organic carbon for measurements of % soil carbon and soil bulk density over the period 1992–2013 [6,40]. The US-Ha1 total soil carbon in the top 15 cm of the mineral soil was 6072 ($\pm$1509), and, 6021 ($\pm$1238) gC $m^{-2}$ for 1992 and 2013, respectively (3; Table S11). The lack of temporal resolution for soil carbon changes in the soil column and the difficulty of sampling for detection of small changes in a relatively large and heterogeneous soil carbon pool are unlikely to yield cost-effective, meaningful, and verifiable data for commercial products, although this approach is commonly employed [23,24]. For example, detection of a significant change ($p < 0.05$) in soil carbon of ~20 gC $m^{-2}$ is estimated to require ~164,194 samples for the US-Ha1 soil sampling study surrounding the hemlock dominated (HEM) tower site (3; Figure S6). In contrast, as described previously, NEE and $R_{eco}$ changes and trends in the flux and the flow of $CO_2$ and isotopocules into and out of the soil matrix establish contemporary and historic perspectives to guide commercial development and conservative practice for soil

carbon sequestration and monetization. NEE integrates complex diurnal and seasonal factors affecting soil $CO_2$ efflux from the soil column extending from the surface to ~1 m in depth including turnover of differing carbon pools, diffusion, short- and long-term changes in temperature, microbial metabolism, root activity and abiotic factors [6]. The US-Ha1 research experiment and similar long-term studies are needed to create and test mitigation and financial approaches to manage climate change with economic, social, and planetary benefits.

The $^{ISO}$NEE products defined for $^{13}C^{16}O_2$ and $^{18}O^{12}C^{16}O$ have the potential to redefine carbon trading dynamics for $CO_2$ measurement derivatives. The carbon and oxygen isotopic species as singular or mixed isotopic masses can be quantified and, as for $^{12}CO_2$, be financialized according to mass across project temporal and spatial domains. The market value of these isotopocules have not been priced but are reasonably expected to trade in multiples of pricing for non-isotopic counterparts. Enhanced data for the partitioning of NEE, not currently available in a commercial forest carbon protocol, will benefit investors and landowners in management of projects in terms of forest operations, financial planning, and carbon isotopic flux trading. The basic approach has been described following from established theory [45,85], methods and field campaigns [16,17,86]. Recognizing the possibilities for $^{ISO}$NEE for $^{14}CO_2$ and isotopocules of $N_2O$ and $CH_4$ are likely to continue to expand carbon trading products based on direct measurement.

We emphasize that extrapolation of areas beyond the US-Ha1 tower footprint presented in this study is not assumed to be representative of the immediate tower forest cover. Extrapolation is employed here solely as a financial projection to illustrate revenue potential across increasing land area for a project. While remote sensing estimates of GPP for the Harvard Forest are similar to that of the surrounding ~16,500 ha [6,87], additional data are required to reliably extend the EC NEE results. For US-Ha1, extension of the NEE landscape would involve additional EC platforms to fill gaps across the selected area (e.g., 16,500 ha), utilize select remote sensing data and employ models to integrate the data. EC network data fused with remote sensing and diverse models are well described including, for example, FLUXCOM [57,65], and subregional networks [88–92]. An encouraging result has been found in the Howland Forest EC site (US-Ho1), where annual mean NEE fluxes between co-located towers were found to be within 5% of each other [93]. Therefore, in a homogeneous terrain, the extrapolation of the results outside the EC area might add an uncertainty of approximately the same order of magnitude as introduced by the gap-filling methodology [34].

## 5. Conclusions

We have shown that the retrospective US-Ha1 research results provide financial terms for an EC project with a projected conservative revenue of at least ~$109 million ($10 tCO$_2$; 40,468 ha; non-exited project), bridging the gap between science and commerce. We suggest that land management for conservation as practiced at the US-Ha1 be expanded across the New England region potentially increasing the $CO_2$ sequestered by ecosystems. Creation of networks of EC towers across the New England region could accurately value and price the contributions of reforestation and conservation projects to mitigate climate change. The retrospective US-Ha1 record implies a complex interplay of biotic, abiotic, and human factors as agents of the magnitude and permanence for forest ecosystems and the carbon they store. The scientific methods described in this study can be applied across global policy platforms such as the Paris Agreement and the UN-REDD Program, harmonizing pricing, and validation for voluntary and compliance markets worldwide.

**Author Contributions:** B.D.V.M. conceived, planned, and prepared the manuscript. N.B. analyzed the data, prepared the graphs and manuscript. J.W.M. provided the Harvard Forest data and contributed to interpretation of the data and writing of the manuscript. All authors have read and agreed to the published version of the manuscript.

**Funding:** This study was not directly funded.

**Institutional Review Board Statement:** Not applicable.

**Informed Consent Statement:** Not applicable.

**Data Availability Statement:** Data archives are available at AmeriFlux (https://ameriflux.lbl.gov/sites/siteinfo/US-Ha1) or through the LTER US-Ha1 site. See: Munger, W. and S. Wofsy. 2020. Canopy-Atmosphere Exchange of Carbon, Water and Energy at Harvard Forest EMS Tower since 1991 ver 31. Environmental Data Initiative. https://doi.org/10.6073/pasta/3367ebb11bb0ef4ddadb928906034351, and for Ameriflux see: J. William Munger (1991-) AmeriFlux US-Ha1 Harvard Forest EMS Tower (US-Ha1R1), Dataset. https://doi.org/10.17190/AMF/1246059. Data Link: https://doi.org/10.17190/AMF/1246059. Isotopic data were obtained from: https://harvardforest1.fas.harvard.edu/exist/apps/datasets/showData.html?id=US-Ha1209.

**Acknowledgments:** We thank Eugene F. Kelly for review of the manuscripts and suggested revisions. External funding was provided by DOE-AmeriFlux and NSF-LTER for operation of the US-Ha1 site. Data archives are available at AmeriFlux or through the LTER US-Ha1 site. See: Munger, W. and S. Wofsy. 2020. Canopy-Atmosphere Exchange of Carbon, Water and Energy at Harvard Forest EMS Tower since 1991 ver 31. Environmental Data Initiative. https://doi.org/10.6073/pasta/3367ebb11bb0ef4ddadb928906034351, and for Ameriflux see: J. William Munger (1991-) AmeriFlux US-Ha1 Harvard Forest EMS Tower (US-Ha1R1), Dataset. https://doi.org/10.17190/AMF/1246059. Data Link: https://doi.org/10.17190/AMF/1246059.

**Conflicts of Interest:** The authors report no conflict of interest.

**Appendix A**

NEE: $R_{eco}$: GPP and 13C RELATIONSHIPS AND TIME PATTERNS

Figure A1 shows NEE, $R_{eco}$, and GPP from the Harvard Forest across different time scales, after filtering and gap-filling. NEE flux is positive (emissions) during night hours and negative (captures) in the daytime because there is no photosynthesis at night and NEE is equal to $R_{eco}$ (Figure A1A). In this case, as NEE is measured and $R_{eco}$ is calculated with Equation (3), these fluxes are not exactly equal at night, while GPP (calculated by difference using Equation (1)) is positive at some intervals. Therefore, this result is biologically incorrect, but it is consistent with the methodology used here for its calculation.

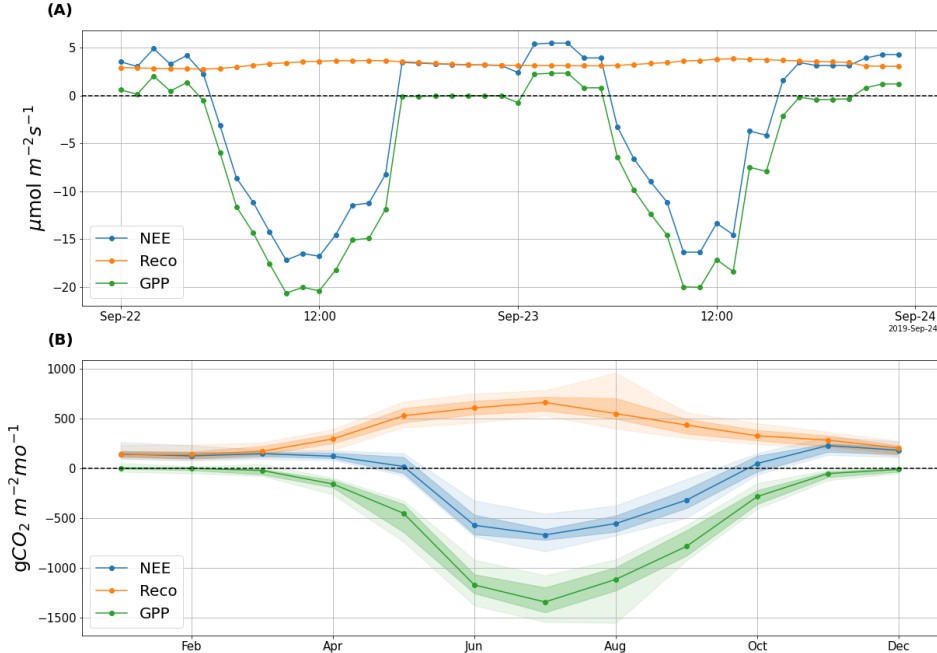

**Figure A1.** From hourly to yearly NEE (blue), $R_{eco}$ (orange), and GPP (green) scales. (**A**) Example of hourly fluxes from 22 September 2019 and 23 September 2019. (**B**) Median values of the three variables (lines), values between the first and third quartiles (heavily shaded areas), and values between the 10 and 90 percentiles (lightly shaded areas) for monthly sums.

In December, January, February, and March, monthly GPP is close to zero and NEE is mostly $R_{eco}$, producing a net emission (Figure A1B), which is coincident with the dormant period of the vegetation when all the leaves from the trees fall and the plants diminish its photosynthesis rate. However, around summer (from May to October approximately), the growing season starts, the forest canopy turns full of green leaves and increases their GPP. This produces net sequestration that compensates the emissions from the other months, resulting in negative NEE values on every year (Figure 1B from the main text). More details about the differences in NEE, $R_{eco}$, and GPP distributions during the growing (weeks 21–40; higher mean and variance) and dormant (weeks 41–19; mean and variance closer to zero) seasons and how they aggregate to produce yearly histograms are illustrated in Figure A2.

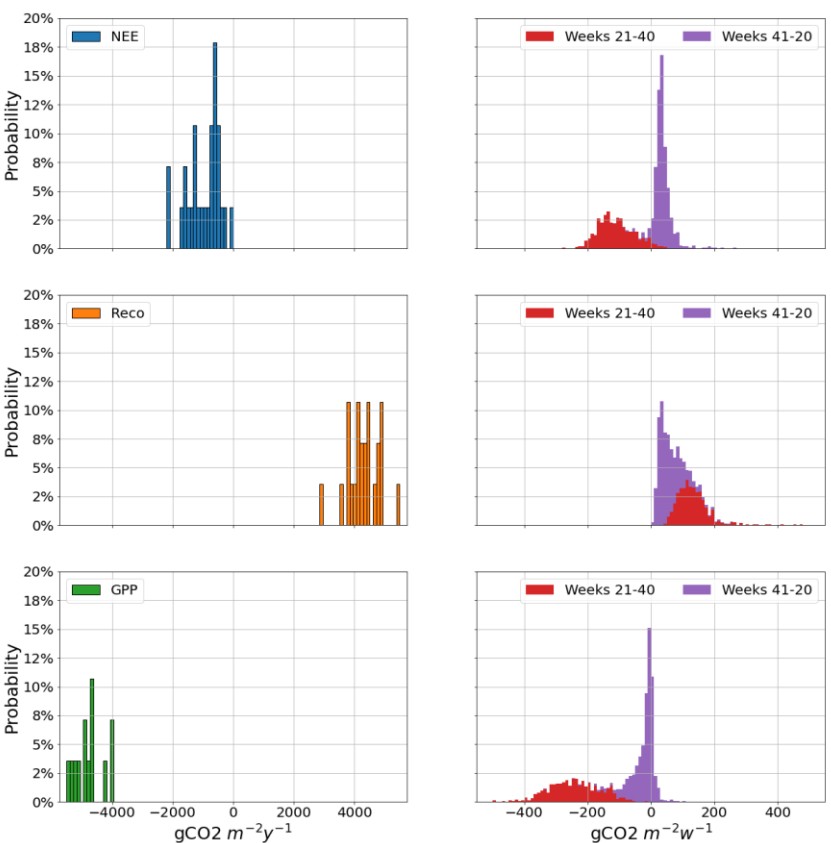

**Figure A2.** Yearly (left) and weekly (right) NEE (top), $R_{eco}$ (middle), and GPP (bottom) histograms. The growing period starts and finishes around week 20 and 40 respectively (red), while the complement is the dormant season (purple).

The relationship between $R_{eco}$ and GPP sums changes each month and each year (Figure A3). However, its correlation is positive, that is, when GPP increases $R_{eco}$ also does, as happened in [94] over all the sites in the Tier 1 FluxNet 2015 dataset. This is because many factors influence both variables in the same way (for example, more leaf area index may result in more photosynthesis, but also more respiration) or are positively correlated (for instance, PAR increases GPP and temperature, which raises the respiration rate). Therefore, it is not possible to increase GPP sums indefinitely without making $R_{eco}$ sums higher.

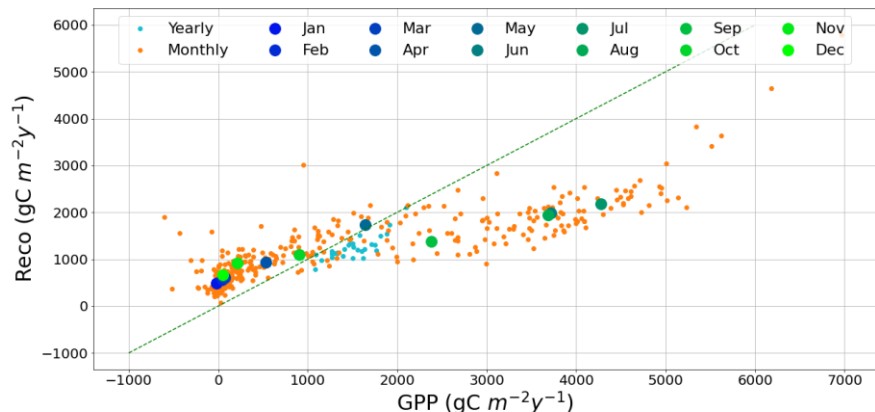

**Figure A3.** $R_{eco}$ vs. GPP for yearly sums (small light blue points), monthly sums multiplied by 12 (small orange points), monthly sums averaged by the month of the year and multiplied by 12 (big points), and the identity line (green dashed line).

**Table A1.** Harvard Forest Annual Statistics.

| | | | | Financial Carbon * | | | |
|---|---|---|---|---|---|---|---|
| | **Physical Carbon** | | | **Extrapolated to Prospect Hill Area (300 ha; 741 ac)** | | **Extrapolated to 40,468 ha (100,000 ac)** | |
| | $gCO_2$ m$^{-2}$ | $tCO_2$ ha$^{-1}$ | $tCO_2$ ac$^{-1}$ | \$10 $tCO_2$ /\$50 t$^{13}CO_2$ | \$30 $tCO_2$ /\$150 t$^{13}CO_2$ | \$10 $tCO_2$ /\$50 t$^{13}CO_2$ | \$30 $tCO_2$ /\$150 t$^{13}CO_2$ |
| No Exit Projects (28-Years) | | | | | | | |
| Total | −27,101 | −271 | −109 | 813,042 | 2,439,126 | 109,722,306 | 329,166,918 |
| Yearly Mean | −978 | −9.78 | −3.96 | 29,358 | 88,074 | 3,961,984 | 11,885,953 |
| Yearly Std | 553 | 5.53 | 2.24 | 17,827 | 53,482 | 2,242,524 | 6,727,602 |
| Min (2010) | −4.21 | −0.04 | −0.02 | 126 | 379 | 17,075 | 51,226 |
| Max (2008) | −2199 | 22 | −8.91 | 65,997 | 197,990 | 8,906,434 | 26,719,302 |
| Exit after 5 years | | | | | | | |
| Yearly Mean | −986 | −9.86 | −3.99 | 1901 | 5705 | 256,522 | 769,657 |
| Yearly Std | 325 | 3.25 | 1.32 | 696 | 2088 | 93,918 | 281,754 |
| Exit after 20 years | | | | | | | |
| Yearly Mean | −1015 | −10.2 | −4.11 | 6610 | 19,832 | 891,776 | 2,675,328 |
| Yearly Std | 65 | 0.65 | 0.26 | 731 | 2193 | 98,642 | 295,928 |
| $CO_2$ Isotopocules ($^{13}CO_2$ or $^{18}O^{12}C^{16}O$) | | | | | | | |
| 2011 | 30.0 | 0.30 | 0.12 | 4502 | 13,506 | 607,620 | 1,822,861 |
| 2012 | 33.4 | 0.33 | 0.14 | 5013 | 15,039 | 676,535 | 2,029,605 |
| 2013 | 44.6 | 0.45 | 0.18 | 2591 | 7772 | 349,660 | 1,048,980 |
| Mean | 36 | 0.36 | 0.15 | 4035 | 12,105 | 544,605 | 1,633,815 |
| Total | 108 | 1.08 | 0.44 | 12,106 | 36,317 | 1,633,815 | 4,901,447 |

* We note that extrapolation of US-Ha1 flux tower data (~1 km$^2$) to landscape scales (e.g., 300 and 40,468 ha) is based solely on numerical projection to illustrate potential project revenue with increasing land area, not an interpretation of ecological net ecosystem exchange for the region.

**Table A2.** Forest sequestration project analysis summary for ~0.92 billion hectares.

| % Area (0.92 Billion Hectares) | Project Length (Years) | Project Area (Millions Hectares) | Average Project Net $CO_2$ Sequestration (Millions) (3 t$CO_2$ ha$^{-1}$ year$^{-1}$) | Annual Revenue (Billions) Carbon Price $10 t$CO_2$ year$^{-1}$ | Project Interval Value (Billions) | Tonne-Year Accounting Exit (Billions) | Project Size (Hectares) | # Projects |
|---|---|---|---|---|---|---|---|---|
| 25% | 5 | 230 | 690 | 69 | 34.50 | 2.25 | 10,000 | 23,000 |
| 25% | 10 | 230 | 690 | 69 | 69.00 | 8.25 | 10,000 | 23,000 |
| 25% | 15 | 230 | 690 | 69 | 103.50 | 18.00 | 50,000 | 4,600 |
| 25% | 20 | 230 | 690 | 69 | 138.00 | 31.50 | 50,000 | 4,600 |
| 100% | - | 920 | 27,600 | 27.60 | 345.00 | 60.00 | - | 55,200 |

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
