# Peer review of "Science to Commerce: A Commercial-Scale Protocol for Carbon Trading Applied to a 28-Year Record of Forest Carbon Monitoring at the Harvard Forest"

_land, doi:10.3390/land10020163_

Round 1

Reviewer 1 Report

The methodolgy in general seems interesting, and the idea that the actual amounts of C sequestered by any given forest mitigation project require better quantification to make them more attractive to investment is certainly true.

Despite this, the paper has serious and significant flaws with respect to underemphasizing the problems and limitations of the study, as outlined in lines 292 to 301.  All modelling studies certainly have their problems and limitations, I don't feel comfortable accepting a paper that seems to try actively to hide them, to the extent that this one does.  For example, the huge financial valuations in Figure 1C are applicable to a hypothetical 40,468 ha landscape under the assumption that the few hectare footprint measured by this flux tower can simply be multiplied by the area, which as we all know is not true.  The hypothetical nature of the landscape is mentioned only in passing in a couple of places and not sufficiently highlighted to avoid misinterpretation.  I could go on with this, but even this one example should be sufficient.

As a result, despite what seems like an otherwise interesting idea and well written paper, I cannot recommend publication.

Reviewer 2 Report

Manuscript ID: Land – 1077770

Science to Commerce: A commercial-scale protocol for carbon trading applied to a 28-Year record of forest carbon monitoring at the Harvard Forest

The manuscript deals with a very important area of science for commercial use. I believe the protocol would be helpful for carbon trading that can be applied not only to carbon monitoring at the Harvard Forest but also to other similar forests having same practices as of the HF. The manuscript is worthy of publishing in the /Land/ journal after having the following major corrections and comments properly addressed.

Abstract:

The abstract of the manuscript jumps directly in the major findings. I think the abstract misses to mention the overall purpose of the study and the investigated research problem, and the basic design of the study. However, the abstract of the manuscript is given with the major findings or trends found as a result of data analysis; and, a brief summary of your interpretations and conclusions.

The abstract should try to avoid abbreviation use. Starting a sentence with an abbreviation should be avoided. I suggest following the suggestion all through the manuscript.

Lines 15-17: The sentence should be revised as “We show that transactions of high frequency direct measurement for CO2 net ecosystem exchange (NEE) track and validate ecosystem carbon dynamics in contrast to model-based 16 approaches that rely on forest mensuration and growth simulations.”

Keywords: Write keywords in alphabetic order. Maintain the superscripts and subscripts where needed otherwise the keyword doesn’t have meaning.

Introduction: The authors start the introduction section with an abbreviation. No sentence can be started with an abbreviation. Follow the advice for the entire manuscript. Sometimes, the introduction section was read as methods. Please, follow the general rules of writing an introduction for a manuscript. The authors mention what they have done in the study but the introduction should be provided with the objective of the study.

This section lacks in sufficient background and should include some more relevant references?

It seems to me that the Introduction section was written for a report, not for a journal article (Lines 71-72).

Lines 37-41: Split the sentence into two or more sentences. It would make it more meaningful and easy to follow.

Materials and Methods:

Reco should be Reco (line 195)

Write details to some extent about the direct measurement of the data.

Results and Discussion: The authors will have to do a major revision here. There are some citations in the results section which should strictly be removed. Otherwise the message(s) of the results section are not getting clear. All discussions in the results should be removed and insert in the Discussion section which therefore should be rewritten.

Lines 366-369: Are the sentences required to give in the Figure title or can be moved to the general statement?

Line 372: (no-exit) or no-exit?

In the Appendix Table A2: Subscript and superscript of some units are not done.

Reviewer 3 Report

Review of manuscript LAND-1077770: Science to Commerce: A commercial-scale protocol for carbon trading applied to a 28-year record of forest carbon monitoring a the Harvard Forest.

This manuscript reports on a long-term study of high quality data on carbon sink and source dynamics at the Harvard Forest and how it can be applied to evaluate incentives for potential private and commercial projects.

I am not a carbon dynamics forest researcher, but am familiar with most of the instrumentation and methods used for data collection. Therefore, I reviewed the manuscript from the perspective of a consulting forester for an individual or company. Overall, I found the manuscript very well written although highly technical, particularly from the very high use of acronyms, which are necessary but tend to slow reading and easy understanding of information provided.

My comments are related to editorial questions:

Line 180: Has PAR been defined?

Line 311: Has "S" in "... the X, S were …" been defined?

Line 312: Is the comma in 86,8 intended or a typo?

Line 326: I believe the California Air Resource Board was defined in line 288 as CARB.

Line 342: Should Harvard Forest use the acronym HF as defined earlier in Line 14?

Round 2

Reviewer 2 Report

The manuscript looks good in the present form.